# Imaging-Guided Delivery of a Hydrophilic Drug to Eukaryotic Cells Based on Its Hydrophobic Ion Pairing with Poly(hexamethylene guanidine) in a Maleated Chitosan Carrier

**DOI:** 10.3390/molecules26247426

**Published:** 2021-12-07

**Authors:** Sofia A. Zakharenkova, Marina I. Lebedeva, Alexandra N. Lebedeva, Irina A. Doroshenko, Ksenya Yu Vlasova, Anastasiya A. Bartoshevich, Vladimir M. Senyavin, Sergey S. Abramchuk, George G. Krivtsov, Alexander A. Ezhov, Tatyana A. Podrugina, Natalia L. Klyachko, Mikhail K. Beklemishev

**Affiliations:** 1Department of Chemistry, Lomonosov Moscow State University, 119991 Moscow, Russia; marinal1807@gmail.com (M.I.L.); spottycucumber@yandex.ru (A.N.L.); doroshenkoiran@gmail.com (I.A.D.); vlasova_k.y@mail.ru (K.Y.V.); nastenabarto@mail.ru (A.A.B.); senyavin@phys.chem.msu.ru (V.M.S.); podrugina@mail.ru (T.A.P.); klyachko@enzyme.chem.msu.ru (N.L.K.); beklem@inbox.ru (M.K.B.); 2Department of Medical Nanobiotechnology, Pirogov Russian National Research Medical University, 117997 Moscow, Russia; 3Laboratory of Physical Chemistry of Polymers, Nesmeyanov Institute of Organoelement Compounds (INEOS RAS), 119991 Moscow, Russia; abr@polly.phys.msu.ru; 4Mechnikov Research Institute for Vaccines and Sera, 105064 Moscow, Russia; invention1938@inbox.ru; 5Faculty of Physics, Lomonosov Moscow State University, 119991 Moscow, Russia; alexander-ezhov@yandex.ru; 6Center for Nanotechnology in Drug Delivery, Eshelman School of Pharmacy, University of North Carolina at Chapel Hill, Chapel Hill, NC 27599, USA; 7Institute “Nanotechnology and Nanomaterials”, G.R. Derzhavin Tambov State University, 392000 Tambov, Russia

**Keywords:** carbocyanine dye, ceftriaxone, cross-linked maleated chitosan, hydrophobic ion pairing, imaging-guided drug delivery, poly(hexamethylene guanidine)

## Abstract

Imaging-guided delivery is developed for hydrophobic drugs, and to a much lesser extent, hydrophilic ones. In this work we have designed a novel strategy for real-time monitoring of hydrophilic drug delivery. Traditionally, the drug and the dye are covalently attached to a nanocarrier or are electrostatically adsorbed. Recently, we found an efficient way to bind the drug by ion-paring with an appropriate counter-ion to form the aggregate that embeds a hydrophobic dye with a considerable fluorescence enhancement. We synthesized a series of carbocyanine dyes of hydrophobicity sufficient for solubilization in hydrophobic ion pairs, which restores their emission in the near-infrared (NIR) region upon the formation of the ternary aggregates. To avoid using toxic surfactants, we applied an amphiphilic polymer-oligomer poly(hexamethylene guanidine) (PHMG) as a counter-ion. Сeftriaxone was used as a model hydrophilic drug ensuring the highest fluorescent signal. The so-formed drug–counter-ion–dye aggregates were encapsulated into a cross-linked maleated chitosan carrier. Confocal laser scanning microscopy (CLSM) studies have demonstrated internalization of the encapsulated model drug by breast adenocarcinoma cells at 40 min after treatment. These results suggest the potential application of hydrophobic ion pairs containing an NIR dye in imaging-guided delivery of hydrophilic compounds.

## 1. Introduction

A fair amount of drug delivery strategies to cells, tissues, and organs has been developed using various nanostructures [1,2]. Less attention was paid to the task of real-time monitoring of delivery [3,4,5,6,7,8]. Imaging-guided delivery of hydrophobic substances is a well-developed strategy, which is based on loading both the drug and a hydrophobic dye (preferably emitting in the infrared (IR) or near-IR region, in one of the tissue transparency windows) into a nanocontainer [9,10] and which allows the drug distribution to be monitored by the fluorescence of the dye until container destruction. Within that strategy, imaging-guided chemotherapy was realized with doxorubicin (DOX) as the ‘gold standard’ of a model drug for cancer treatment using various nanocontainers: cross-linked chitosan with carbon dots as fluorophore [11], *N*-naphthyl-*O*-dimethymaleoyl chitosan-modified magnetic nanoparticles with DOX as fluorophore [12]; a self-assembled glutathione-responsive porphyrin with DOXas fluorophore [3] and β-cyclodextrin–carbocyanine conjugate [4] were used to deliver DOX to HeLa cells. Paclitaxel was packed in magnetite–polystyrene–poly(lactic-co-glycolic acid) nanospheres with NIR quantum dots as fluorophore to treat tumors in mice [13].

As for hydrophilic drugs, they can easily penetrate the cell membranes without the use of any specific delivery vehicles. To neutralize the charge on an ionic drug, the surfactant-like counter-ions are applied to form hydrophobic ion pairs (HIPs) producing hydophobic complexes with the drug via electrostatic interactions [14]. HIPs are widely used in drug delivery [15,16] but not in imaging. Delivery systems are also used for the hydrophilic drugs that show poor bioavailability through conventional routes of administration; those types of drugs can be delivered using nanoemulsions applied to the skin [17] or orally [18]. However, monitoring of the delivery of hydrophilic substances has been much less studied. Monitoring of this type of compounds can be based on covalent or non-covalent labeling of the container. Covalently attached fluorescent label [5,6] indicates the position of the nanocontainer, but does not reveal its destruction. A more promising approach is reversible non-covalent labeling of the drug [7,8,19], which not only eliminates the covalent bond formation, but preserves the chemical identity of the delivered compound and also makes it possible to observe the destruction of the container upon drug release. There are few examples of using hydrophobic ion pairing or self-assembly for imaging of hydrophilic drugs: nicardipine was imaged by quenching an anthracenic dye in the chitosan nanocontainer and regaining fluorescence upon drug release [8]; cysplatin prodrug was electrostatically linked to carbon dots, incapsulated into a polymeric shell [7]; a multifunctional nanocarrier containing an SN-38 antitumor agent was assembled with a carbocyanine dye [19]. However, too rapid release of a drug attached by adsorption can present a problem of non-covalent drug binding [20]. A more general approach to strong non-covalent binding would be well-suited for monitoring the hydrophilic drugs.

Hydrophobic ion pairing is an appropriate type of binding for a hydrophilic drug when its imaging is required. As we have recently shown [21], HIPs (for example, formed by cephalosporin anion and cetyltrimethylammonium cation) can incorporate a hydrophobic dye with a considerable fluorescence enhancement. We synthesized [21] a new series of pentamethine carbocyanine dyes with appropriate hydrophobicity to be solubilized in the HIPs. In aqueous buffer solution, the dye exists as nanoparticles, and its fluorescence is quenched (aggregation-induced quenching [22]). The bright emission in the near-IR region is restored when the dye enters the hydrophobic domains of the HIPs formed by the alkyl chains of the counterion (for example, cetyltrimethylammonium). We consider that these systems could be suitable for non-covalent imaging of a hydrophilic drug as a component of the ion pair. The incorporated dye-containing fluorescent HIPs (“ternary aggregates” [21]), when placed in an appropriate nanocarrier, may indicate the position of the drug in the organism. Moreover, it will be possible to observe the nanocontainer lysis that might be predictably accompanied by a change in the quantum yield of the IR fluorophore upon contact with the internal milieu.

For the transfer of the ternary aggregates across cell membranes, we obtained their complexes with anionic chitosans, widely used as a drug delivery vehicle. Chitosan is regarded as a biocompatible, biodegradable, and non-toxic biomaterial, easy modifiable and useful as a drug carrier [23,24]. Anated chitosans are considered promising in drug delivery due to their lower toxicity and favorable permeation properties. Carboxymethylated chitosan is one of the best studied [25], although less attention has been paid to other anionic chitosans [26]. In this work, we evaluated the possibility for formation of nanocarriers containing fluorescent hydrophobic ion pairs (ternary aggregates), which carry an NIR fluorescent dye, using both maleated and carboxymethylated chitosan cross-linked with dialdehydes (see Figure 1 for the structures and Figure 2 for the overall diagram of the processes). Such hydrophilic containers were regarded unsuitable for imaging-guided delivery based on the quantum yield change of the dye, since it was difficult to encapsulate the hydrophobic dye into a hydrophilic container and because water penetrated the container rapidly with dye emission quenching [22]. In our work both these difficulties have been overcome by placing the dye into the novel ternary aggregate drug–counter-ion–dye, which was then encapsulated into anated chitosan. The aggregate itself is hydrophilic and only contains the hydrophobic domains intended to hold the dye.

The purpose of this study is to assess the feasibility in application of the dye-containing chitosan-encapsulated ternary aggregates for the delivery of a hydrophilic drug to eukaryotic cells. We used ceftriaxone as a model drug, which is known to form [21] stable ternary aggregates with CTAB and capable of fluorescence enhancement of a pentamethyne carbocyanine dye. We replaced the toxic surfactant (CTAB) used as a counter-ion in [21] with a cationic polymer-oligomer poly(hexamethylene guanidine) (PHMG). This compound is prepared by polycondensation of hexamethylenediamine and guanidine hydrochloride that yields weakly branched oligomers of a low molecular weight (up to 2 kDa, or a polymerization degree of 10–15) with amino or guanidine as terminal groups [27]. The pKa of protonated PHMG is 12.5, which makes it a polycation over a wide pH range (up to 12) [28], which is suitable for obtaining hydrophobic ion pairs with the drug. PHMG is known to be a potent antimicrobial agent: its 1 ppm aqueous solution exhibits an antibacterial activity above 90% [27]. Secondly, PHMG has low toxicity to mammals as it has been shown that their enzymes efficiently decompose the compound (LD(50)), and PHMG was found to be at a concentration 600 mg/kg when administered as a single dose via the stomach tube [29]. To obtain healing and antimicrobial films, PHMG was used as complexes with chitosan [30] and polyvinyl alcohol [30,31].

As a result, we have found that the chitosan containers containing the aggregates (drug–PHMG–carbocyanine dye) display a stable NIR fluorescent signal and can be uptaken by breast adenocarcinoma cells. In the literature, there are few examples of utilization of ion pairing for non-covalent labeling of a drug for its imaging-guided delivery [7,8,19]. An important feature of the suggested system is that the quantum yield is dependent on the fluorophore environment, namely the drug–counter-ion aggregate (in molecular form) exhibits strong fluorescence whereas the dye emits weakly when present in a nanoparticle form in an aqueous solution. This property is supposed to enable real-time monitoring of container destruction with the liberation of the drug. The use of fluorescence quenching on container decomposition [22] is an emerging trend in delivery systems. For instance, it was used in [8] but in that paper hydrophobic interactions were used rather than ionic self-assembly, which does not allow for binding the hydrophilic drugs. In papers [7,19], ionic self-assembly was utilized to bind the drug and the fluorophore, but the latter was not supposed to change its emission intensity upon container destruction. Overall, in this work we are using a unique combination of delivery system attributes.

## 2. Results and Discussion

### 2.1. Aggregation Studies in Poly(hexamethylene guanidine)–Ceftriaxone–Dye System

#### 2.1.1. Quenching of the Dye Fluorescence in Water

Fluorescence enhancement in the ternary aggregates poly(hexamethylene guanidine)–ceftriaxone–dye is only observed upon pre-quenching of the fluorescence of carbocyanine dye in water. The dye is emissive in organic solvents, but upon mixing with an excess of aqueous buffer it forms a colloidal solution of sphere-shaped nanoparticles [21], which absorb and emit very weakly (Figure 1).

#### 2.1.2. Aggregation in the Ternary System PHMG–Ceftriaxone–Dye

In binary systems (dye–ceftriaxone and dye–PHMG), no emission enhancement is observed. Ceftriaxone is negatively charged (at physiological pH of 7.4 used in study, its charge is –1 [32]), while PHMG exists as a polycation over a wide pH range [28]. Ceftriaxone and PHMG form ionic aggregates containing hydrophobic domains due to PHMG alkyl chains and are capable of solubilizing the carbocyanine dye nanoparticles, similar to the ceftriaxone–cetyltrimethylammonium (CTAB) system studied in paper [21]. Such solubilization is accompanied by a considerable fluorescence enhancement (the spectrum is shown in Figure 2b). Moreover, Rayleigh light scattering spectrum of the ternary system (Figure 2a) shows a notable difference between the spectra of the binary mixtures and single components, confirming the formation of ternary aggregate PHMG–ceftriaxone–dye.

To estimate stability constant of the aggregate formed, the concentration dependences were studied for the system PHMG–ceftriaxone–dye (Figure 3). The emission intensities of binary systems (blue curves) are close to that of the background signal of the cell lacking a fluorophore, as measured by the NIR visualizer. On these grounds, the signal was postulated to be exclusively due to the emission of the ternary aggregates. As seen in Figure 3, NIR fluorescence intensities are represented as complex curves. However, their portions can be used for an estimation of the aggregation stability constant by using the Benesi–Hildebrand method [33]. The simplest complexation scheme can be written as the following equilibrium reaction:
P + C + D = РCD,
where P is PHMG, C is ceftriaxone, D is dye, and the molar concentration of PHMG is calculated with respect to its monomer unit. The stability constant was calculated according to formula (see Appendix B for the details):$$K=\frac{\left[\mathrm{PCD}\right]}{[P][C][D]}$$

The obtained values of the constant for PHMG and CTAB as counter-ions determined from two different curves (Table 1) should only be considered as rough estimates due to the complexity of the system. The value of ~10^6^ (or about 10^4^ if recalculated for two interacting particles) is comparable by the order of magnitude with the stability of ion pairs of an organic cation (celiprolol) with chloride ion (5 × 10^5^ [34]) or an organic anion citrate(3–) with Ca^2+^ cation (1.5 × 10^3^ [35]), which confirms moderate stability of the aggregates and their ability to decompose on dilution. The stability of the PHMG aggregate is five-fold higher than that of CTAB, most probably due to the polymeric nature of PHMG.

### 2.2. Encapsulation of Ternary Aggregates in Chitosan Containers

#### 2.2.1. Anated Chitosans Used in Developing Containers

To assemble delivery vehicle, the ternary aggregates of ceftriaxone–PHMG–dye were encapsulated into biocompatible containers, which have been prepared using the common types of anionic chitosan, namely maleated and carboxymethylated, obtained according to known methods [36,37]. Carboxymethylation of chitosan with sodium chloroacetate was supposed to occur predominantly at O-6 [38], whereas the reaction with maleic anhydride leads to carboxymethylation of the amino group [39]. Additionally, sulfated maleated chitosan was obtained from maleated chitosan by treatment with metabisulfite (a source of hydrosulfite) as a result of reaction of the sulfonic acid group with the maleate double bond [37].

The ceftriaxone–PHMG–dye–chitosan complexes were obtained by precipitation from the mixture of the aqueous solutions of the components at physiological pH 7.4 (phosphate buffer).

#### 2.2.2. Chitosan Cross-Linking

Dialysis studies (data not shown) indicated that uncrosslinked quaternary complexes decompose in less than 1 h and become non-fluorescent. To overcome that disadvantage, the chitosan containers were cross-linked with aldehydes, which are known to increase stability of containers [40]. Glutaraldehyde was among the most efficient cross-linkers [41]. We used formaldehyde, glyoxal, and glutaraldehyde according to the following procedure. The aldehyde solution was added to the ultrasound-pretreated suspension of containers at pH 7.4 and left for a day at room temperature. The resulting suspension was sonicated following by fluorescence monitoring. Only the containers, which have been obtained using glutaraldehyde, showed measurable fluorescence (Figure 4). Since this aldehyde has been found to be a good cross-linking reagent for carboxymethylated and sulfated maleated chitosan, it was considered as a potent cross-linker in further work.

### 2.3. Characterization of Poly(hexamethylene guanidine)–Ceftriaxone–Dye–Maleated Chitosan Containers

#### 2.3.1. Morphology and Size of Container Particles

TEM images showed the cross-linked chitosan containers to be nanoparticles of about 70 nm in diameter in the form of chains and conglomerates (Figure 5a). The TEM image of similar containers obtained without ceftriaxone indicated that the nanoparticles are somewhat smaller (40–50 nm), but otherwise their morphology is similar to that of the containers containing ceftriaxone (Appendix A). In the absence of chitosan, the ternary aggregates represent 100–300 nm sphere-shaped particles with an electron-dense core, similar to the particles of the dye (Figure 1c).

The particle size of cross-linked systems estimated by dynamic light scattering (DLS) technique (Table 2) has shown to be predictably increased for chitosan–PHMG compared to that for the quaternary system PHMG–ceftriaxone–dye–chitosan. A slight reduction in size observed for maleated chitosan in contrast to that for chitosan–PHMG can be explained by the polyelectrolyte coil shrinking upon interaction of anionic chitosan with PHMG cations. According to DLS data, the obtained containers are polydispersed in aqueous solution due to aggregation processes having particle size varying from the smallest aggregates of 200–300 nm in diameter (by the number distribution, Table 2) to the largest ones of micron size (by the intensity distribution, Table 2). Particles of those sizes are visible in TEM. The complete size distributions provided by Zetasizer software are given in Appendix A.

#### 2.3.2. Zeta Potentials

Since the positive charge of the carrier can lead to more efficient uptake by cells, the molar ratio of PHMG to ceftriaxone was maintained at 2:1. The zeta potentials of particles for the ceftriaxone–PHMG–dye–chitosan system are shown in Figure 6. The dye nanoparticles have no intrinsic charge (ξ = +1 mV), but they acquire it by adsorbing PHMG (ξ = +23 mV). The ternary aggregates PHMG–ceftriaxone–dye are also positively charged (+17 mV) in the excess of PHMG, which allows them to interact with anionic chitosan. Among all crosslinked chitosan-containing particles, which retain the positive charge imparted to them by PHMG (+24 … +28 mV), the target containers chitosan–ceftriaxone–PHMG–dye hold out the prospect of efficient endocytosis. The obtained values also indicate that the nanoparticles can be stable on storage. In principle, the charge can be controlled by changing the PHMG:ceftriaxone ratio.

#### 2.3.3. FT-IR Spectra

Unmodified chitosan (Figure 7a) shows major peaks at 3280/3360, 2860–2930, 1660, 1593, 1423, 1377, and 1000–1150 cm^−1^, corresponding to ν(OH/NH), ν(CH), ν(C=O), δ(NH_2_), two δ(COH), and mixed ν(C-O-C/C-OH) vibrations, respectively [42]. After maleation and cross-linking (Figure 7b), the chitosan peaks of δ(NH_2_) and ν(C=O) at 1593 and 1660 cm^−1^ are shifted to spectral regions of lower energy at 1555 and 1633 cm^−1^, respectively, which agrees with the data described in [43] wherein 22–36-cm^−1^ shifts of these bands were also observed. A peak from remaining acetamide group of chitosan ν(C=O) is present at 1656 cm^−1^. The 1633 cm^−1^ peak also corresponds to the amide group δ(NH_2_) of chitosan, which resulted from cross-linking with aldehydes [44]. A shift of the δ(COH) peak was detected from 1377 to 1356 cm^−1^ upon maleation, and an increase in intensity of the δ(COH) peak at 1423 cm^−1^ was observed. Appearance of the peak at 1746 cm^−1^ has confirmed the formation of carboxylic group due to maleation of chitosan, which was found to be similar to that for carboxymethylated chitosan (Appendix A).

The mentioned above peaks are also present in the maleated chitosan containers sample (Figure 7c). The absorbance of containers is increased due to the presence of ceftriaxone and PHMG. The FT-IR spectra for the containers show absorption at 1352, 1415, 1465, 1633 cm^−1^ corresponding to the unbound ceftriaxone peaks at 1370, 1395, 1500, 1655 cm^−1^ [45]. Similarly to the intense δ(NH) band of unbound PHMG at 1635 cm^–1^ and a δ(CH_2_) peak at 1460 cm^–1^ reported in paper [46], we detected the peaks at 1633 and 1465 cm^–1^, respectively, in the container spectrum (Figure 7c). There was no band specific of dye in the IR spectra due to its low amount in the containers (0.2% mass).

The IR spectra of other functionalized chitosans confirming their chemical modifications are given in Appendix A.

#### 2.3.4. Kinetic Stability

The fluorescence spectra of container suspension, which has been stored at 4 °C, were recorded at one-week intervals. The stability of the NIR signal has been observed for at least a week (Figure 8). A similar result was obtained by Mendes et al., who obtained chitosan nanoparticle stability during a week under the same conditions [47]. The other containers were stable up to 14 days, their size and surface charge changing slightly during that period [48].

#### 2.3.5. Drug Loading Capacity

Drug loading capacity (DLC) was estimated as follows:$$\mathrm{DLC}=\frac{c\left(\mathrm{initial}\right)-c\left(\mathrm{solution}\right)}{c\left(\mathrm{container}\right)}100\%$$

Here, *c*(container) is the mass concentration of container material, *c*(initial) is the mass concentration of ceftriaxone introduced in the solution during the preparation of the containers, and *c*(solution) is the concentration of ceftriaxone in the supernatant obtained after centrifugation of the cross-linked container suspension in PBS. The following formula was used: *c*(solution) = *A*/*ε* × *f*, where *A* is absorbance at 241 nm, *ε* is molar absorption coefficient of ceftriaxone (2.4 × 10^4^), and *f* is dilution factor (*f =* 125). The spectrum was recorded with respect to the blank solution of container without ceftriaxone. The same calculation was performed using a ceftriaxone absorption band at 278 nm and the results were averaged to obtain the value of *c*(solution) = 0.038 mg/mL. Given *c*(initial) for ceftriaxone was 1 mM (0.66 mg/mL), *c*(solution) = 0.62 mg/mL and concentration of the suspension was 0.54 mg/mL with respect to chitosan, the estimate of ceftriaxone loading was 7% by mass. The obtained value is moderate but it is sufficient for the proof-of-concept study.

### 2.4. Cytotoxicity Measurements

According to the ISO 10993-5 standard, the threshold of cytotoxic biomaterial is below 70–80% cell viability [49]. Cytotoxicity test by WST-1 assay (Figure 9) showed that 70–80% cell viability following 24 h of exposure to chitosan containers was observed at the container concentration below 0.2 mg/mL, which is sufficient for in vitro or in vivo imaging. According to the tests of container components, the main contribution to the final formulation cytotoxicity was made by PHMG: this compound reduced the cell viability stronger than the containers. These data are in a line with literature [50]. Aqueous ceftriaxone solutions at the concentrations of up to 0.5 mM have shown low cytotoxicity, which is consistent with the literature [51]. The solutions of chitosan and dye exhibited cytotoxicity, which appeared to be less or comparative to that of the containers. Overall, we found the suggested containers were non-toxic, as was observed in papers [52,53] for the other carboxymethylated chitosan particles.

### 2.5. Chitosan Containers Uptake by Breast Adenocarcinoma Cells

The cells were incubated with the chitosan containers in confocal microscopy plates (40 min, 37 °C) with the subsequent fixation with formaldehyde. For these experiments, all chitosans used for obtaining containers were labeled with Rhodamine isothiocyanate [54]. The conditions of fluorescence measurement in CLSM are given in Table 3.

As shown in Figure 10, NIR fluorescence of both dye and Rhodamine (which was used as a label for chitosan), is visible in the cytoplasm and nucleus of the cell. The container components (dye, ceftriaxone, and chitosan) are capable of entering the cells separately (Appendix A). The emission intensities of Rhodamine that is proportional to the amount of chitosan (channel 1) are approximately equal in all systems, suggesting that chitosan particles can enter the cell either being attached to the drug or in the free form.

In this study, it is pertinent to address the problem of understanding whether the fluorescence of the container delivered into the cells is caused by incorporation of the nanocarrier or is a result of a free dye uptake. Given there is a difference in the intensities of the signals for the loaded and non-loaded carriers, similar to that described in literature [55,56], we can assume that the cells uptake the container. To test whether the NIR signal (channel 2) belongs to the remaining intact quaternary container dye–PHMG–ceftriaxone–chitosan, we have calculated the colocalization coefficients of both dye (channel 2) and chitosan (channel 1). The data obtained showed that they are in the range of 0.83–0.91 (Figure 10), which confirms that in the cell, the dye and chitosan are located in the same area (probably in the form of containers).

Besides, we have compared the NIR intensities (channel 2) for the containers with ceftriaxone and those without it (Figure 10). The latter, which presented less bright emission, indicated the lack of ceftriaxone–PHMG aggregates, which could solubilize the dye, thus causing NIR emission enhancement (channel 2; a quantitative comparison of intensities is shown in Figure 11). For carboxymethylated chitosan, there was no spectral intensity differences observed for the system containing the drug and without one. Overall, these data confirm that fluorescence manifested by the cells is caused by entire containers rather than their individual components. We suggested that ceftriaxone is actually delivered to the cells in the form of a ternary aggregate encapsulated in maleated chitosan.

## 3. Materials and Methods

### 3.1. Compounds

The carbocyanine dye was synthesized according to [21] (see Appendix A). Chitosan obtained from Bioprogress (Shchelkovo, Russia) had a viscosity average molecular weight of 300 kDa, polydispersity index of 0.65 and deacetylation degree of 85%. PHMG with an average polymerization degree of 11 was received from Institute of Ecotechnologies (Moscow, Russia). Other reactants were purchased from Sigma–Aldrich (Taufkirchen, Germany) and used as received. Acetate (pH 3–5), phosphate (pH 6–8), and borate (pH 9–10) buffer solutions were used to maintain pH values. Phosphate buffered saline (PBS) was prepared by dissolving 1.6 g NaCl, 0.04 g KCl, 0.288 g Na_2_HPO_4_ and 0.049 g KH_2_PO_4_ in 100 mL of water. Millipore water (18 mΩ·cm) or 95% ethanol (Bryntsalov-A, Moscow, Russia) were used in preparing solutions.

### 3.2. Instrumentation

Fluorescence and Rayleigh light scattering (RLS) spectra were obtained using a “Fluorat-02 Panorama” spectrofluorometer (Lumex, Saint Petersburg, Russia) in 1-cm length quartz cells. The UV-vis absorption spectra were recorded on SF-102 spectrophotometer (Interphotophysica, Moscow, Russia) in 1-cm quartz cells. Near IR fluorescence in 96-well plates (Thermo Scientific Nunc F96 MicroWell, white, cat. No 136101, Thermo Fisher Scientific, Waltham, MA, USA) was registered using a setup [21] containing an LED source (eleven 3-Wt red LEDs, emission maximum 660 nm; Minifermer, Moscow, Russia) and an NIR digital camera—modernized Nikon D80 (Nikon, Tokyo, Japan) with a filter transmitting light with wavelengths above 700 nm. Particle size distribution by the dynamic light scattering (DLS) technique and zeta-potentials were measured with a Zetasizer Nano ZS (Malvern Panalytical, Malvern, UK). FT-IR spectra were obtained using a Tensor 27 (Bruker, Bremen, Germany) spectrometer. Electronic micrographs in a transmission electron microscope (TEM) were obtained by applying the investigated solution onto a standard TEM copper grid covered with Formvar film (thickness 50 nm), drying on air, and imaging with a LEO 912AB OMEGA transmission electron microscope (Carl Zeiss, Oberkochen, Germany) with accelerating voltage 100 kV. Fluorescence images were obtained by confocal laser scanning microscope (CLSM) Olympus FluoView FV1000 (Olympus, Tokyo, Japan) equipped with a spectral version scan unit based on a motorized inverted microscope Olympus IX82 (Olympus, Tokyo, Japan). The 40 × objective lens with a numerical aperture of 0.9 was used in the measurements. A Sonopuls ultrasonic homogenizer (Bandelin, Berlin, Germany) was used to disperse suspensions. Dialysis tubes (Pur-A-Lyzer Maxi 3500 or Midi 3500, MWCO 3.5 kDa, Sigma, Taufkirchen, Germany) were used for dialysis.

### 3.3. Preparation of Maleated Chitosan

Maleated chitosan was prepared according to [37]. A total of 0.90 g of chitosan was dissolved in 90 mL of 0.2 M acetate buffer (pH 5.0) with stirring for a few hours. After complete dissolution, 1.8 mL of 3 M maleic anhydride in dioxane was added, and the obtained precipitate was dissolved by adjusting the pH value to 8.7 with 9.5 mL of 3 M KOH solution. Next, another 1.8 mL portion of maleic anhydride was added to the resulting solution and the product was neutralized with 3 M KOH to pH 8. The final concentration of chitosan in solution was 0.045 M by repeating unit.

The amount of free amino groups remaining after maleation was determined by using the trinitrobenzene sulfonic acid (TNBS) method [57]. Briefly, aliquots containing 0.5–1.0 mmol chitosan (by repeating unit) or 0.1–1.0 mmol L-leucine as standard were mixed with 500 µL of 0.067 M freshly prepared aqueous solution of TNBS (Sigma), an equivalent of KOH (330 µL of 0.1 M solution), 500 µL of 0.05 M borate buffer (pH 8.5) and water up to the total volume of 2.0 mL and heated during 1 h in a thermostat at 50 °C. The absorbances of the resulting solutions were measured at 340 nm. Counting the number of free amino groups, which remained to be intact after maleation, gave the value of 29% that was considered to be sufficient for cross-linking.

### 3.4. Preparation of Sulfated Maleated Chitosan

Sulfated maleated chitosan was prepared according to [37]. A portion of 10 mL of the solution obtained under 3.3 was brought to pH 4–5, 0.188 g of sodium metabisulfite powder was added in portions with stirring, and the solution was heated at 50 °C for 3 h, maintaining the pH within 4–5 by adding 1 M sulfuric acid. The solution was centrifuged at 2750 rpm for 3 min, the upper layer was separated and dialyzed against water for 19 h.

### 3.5. Preparation of Carboxymethylated Chitosan

Carboxymethylated chitosan was prepared according to [36]. A total of 800 mg of chitosan was dissolved in 80 mL of 0.5% HCl and stirred for a day at room temperature for dissolving. Ten milliliters of this solution was mixed with 10 mL of 3 M KOH dissolved in ethanol:water (2:1, *v*/*v*) while stirring. The precipitate of chitosan was centrifuged at 2750 rpm for 2 min and then washed with water. The chitosan residue was dispersed in water with added 0.7 g of sodium chloroacetate (total volume 6 mL, pH 13) and placed on an orbital shaker for a day. The suspension was neutralized to pH 8, filtered and acetone was added until the formation of a white precipitate, which was separated, washed with a 3:1 mixture of methanol/water and dissolved in 7 mL of 1.2 mmol of acetic acid to obtain a solution with pH 3.8.

### 3.6. Labeling of Chitosans with Rhodamine B

All types of anated chitosan were modified with Rhodamine B isothiocyanate (RITC) according to [45]: 2,0 mL of ethanolic RITC (0.7 mM) solution was added to a 0.045 M modified chitosan solution (2 mL) and kept for a day at room temperature. Then, the mixture was dialyzed against water for two days with several changes of the external solution.

### 3.7. Preparation of Containers Dye–PHMG–Ceftriaxone–Chitosan

The solutions: 0.0056 M PHMG (6 mL), 0.1 g/L dye (380 µL) and 0.005 M ceftriaxone (3 mL) were mixed in a plastic test-tube, and 450 µL of 0.045 M solution of an anated chitosan (or 900 µL of 0.022 M chitosan labeled with RITC) was added. The formed precipitate was centrifuged at 10,000 rpm for 2 min and washed with water. Next, 50% glutaralehyde (220 µL) and water to the final volume of 15 mL were added. The pH of the solution was adjusted to 7.0–7.3 with 1 M KOH and the suspension was stored at room temperature for 24 h for cross-linking. Then, the containers were ultrasonically dispersed and dialyzed against water for 30 min.

### 3.8. Cell Culture and Endocytosis

MCF7 (ATCC^®^ Cat. No. HTB-22™) breast adenocarcinoma cells were cultivated in Thermo Scientific™ Nunc™ Lab-Tek™ II chambered coverglass cells in DMEM medium (Gibco, Paisley, UK) supplemented with 10% of fetal bovine serum (FBS, Gibco, UK), pyruvate (Gibco, Paisley, UK), glutamine (Gibco, Paisley, UK) and antibiotics and antimycotic (Gibco, Paisley, UK) at 37 °C with 5% CO_2_. To every chamber, 300 µL of cross-linked quaternary container dye–PHMG–ceftriaxone–chitosan was added. Cells were incubated at 37 °C with 5% CO_2_ for 40 min, the solution was removed, cells were washed with PBS and fixed with 4% formaldehyde in PBS for 15 min. A total of 500 µL of conservation mixture glycerol:PBS (1:1 *v*/*v*) was added to every chamber, and the samples were imaged in CLSM.

### 3.9. Cytotoxicity Measurements

MCF7 cells were seeded in a 96-well plate (5 × 10^3^ cells/well) in 200 µL/well of complete DMEM medium and incubated at 37 °C with 5% CO_2_ for 48 h. Then the medium was replaced with 100 µL/well DMEM medium in the absence (control) or presence of various amounts of the tested compounds and incubated at 37 °C with 5% CO_2_ for 24 h; 10 µL/well WST-1 solution (CELLPRO-RO Roche, Basel, Switzerland) was added to each well and incubated under culture conditions for 2 h. The absorbance of the samples was measured at 450 nm.

## 4. Conclusions

We suggested a feasible approach for monitoring of the delivery of large hydrophilic compounds into eukaryotic cells. The technique is based on the aggregate formation of the delivered compound linked with an oppositely charged ion and a hydrophobic dye, followed by its encapsulation into a polyelectrolyte container. The strategy is demonstrated using the ceftriaxone–PHMG–carbocyanine dye system in the containers of dialdehyde-crosslinked maleated chitosan, penetrating across the membrane of human breast adenocarcinoma cells. The containers are shown to have low toxicity and high temporal stability.

As a prospect for the future, other potent systems should be developed that would allow the difference between the bound and free drug in the cell to be clearly distinguished in order to observe the in vivo degradation of containers in the course of time. We consider such methodology will pave the way for real-time imaging-guided delivery of hydrophilic drugs.

## Data Availability

Not applicable.

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
