# Peer review of "Imaging-Guided Delivery of a Hydrophilic Drug to Eukaryotic Cells Based on Its Hydrophobic Ion Pairing with Poly(hexamethylene guanidine) in a Maleated Chitosan Carrier"

_molecules, 2021, doi:10.3390/molecules26247426_

Round 1

Reviewer 1 Report

I hope you're well.
The article entitled Imaging-Guided Delivery of a Hydrophilic Drug to Eukaryotic Cells Based on its Hydrophobic Ion Pairing with Poly(hexamethylene Guanidine) in a Maleated Chitosan Carrier was submitted to the journal Molecules. The work is well written, the experiments are substantiated, and the results are clear and precise. Just a few small considerations in reviewing the English language.

Author Response

We greatly appreciate the time and comments from the Reviewers. They are very helpful for improving our paper.

The changes in the revised version of manuscript are shown in track changes mode. The minor language edits are numerous and have not been tracked to avoid distracting the reader’s attention.

Point 1: Just a few small considerations in reviewing the English language. 

Response 1: The language has been carefully edited with the help of an expert.

Reviewer 2 Report

Dear Authors,

The manuscript entitled "Imaging-Guided Delivery of a Hydrophilic Drug to Eukaryotic

Cells Based on its Hydrophobic Ion Pairing with Poly(hexamethylene Guanidine) in a Maleated Chitosan Carrier” is a very interesting and pertinent topic.

Please, consider the following suggestions to improve your manuscript:

  1. Describe the meaning of the abbreviations, such as IR and NIR.
  2. Page 2:

cross-linked chitosan (carbon dots as fluorophore) [11], N-naphthyl-O-dimethymaleoyl chitosan-modified magnetic nanoparticles (DOX as fluorophore) [12]; a self-assembled glutathione-responsive porphyrin (the same as fluorophore) [3] and β-cyclodextrin–carbocyanine conjugate [4] were used to deliver DOX to HeLa cells. Paclitaxel was packed in magnetite–polystyrene–poly(lactic-co-glycolic acid) nanospheres (near-IR quantum dots as fluorophore) to treat tumors in mice [13].”

Reword the text.

Avoid excessive parentheses. It gets confusing.

  1. Page 2:

“Placed in an appropriate nanocarrier, the fluorescent HIPs containing the incorporated dye (we name them "ternary aggregates" [21])”

Remove “we name them”

  1. Page 3:

“The purpose of this study was to demostrate”

Correct - demonstrate

  1. Remove references in subtitle (ex. 3.3 and 3.4), include references in the text.
  2. Figures 10 and 11 should be included in the results section, on page 12.
  3. Explain further how it was calculated the “Drug loading capacity”
  4. Why did you use the WST-1 assay to assess cytotoxicity? Instead of MTT?
  5. I suggest adding a section with the conclusions obtained.

Author Response

We greatly appreciate the time and comments from the Reviewers. They are very helpful for improving our paper.

The changes in the revised version of manuscript are shown in track changes mode. The minor language edits are numerous and have not been tracked to avoid distracting the reader’s attention.

For our replies please see the attachment.

Reviewer 3 Report

The topic proposed by the authors is interesting.

The idea is interesting but the paper should be improved before to be accepted for publication in this journal.

After reading the manuscript, the following doubts and suggestions have arisen.

  • the introduction should be more complete, providing supplementary background about the various properties of poly(hexamethylene guanidine.
  • the authors should mentioned the polydispersity index and average gravimetric mass of the chitosan used.
  • missing information about the manufacturer of NIR digital camera (name, city, country) and confocal laser scanning microscope.
  • there is a lack of greater comparisons of the results obtained in this experiment with other similar studies, and preferably in the last 5 years.
  • there is other supplementary information in the literature about the various designs of chitosan-polyvinyl alcohol/polyhexamethylene guanidine systems and their use, especially for antibacterial effects. Literature analysis reveals multiple communicated data in this field, that should be introduced in the discussion section:
  • Yue X, Liu L, Wu Y, Liu X, Li S, Zhang Z, Han S, Wang X, Chang Y, Bai H, Chai J, Hu S, Wang H. Preparation and evaluation of chitosan-polyvinyl alcohol/polyhexamethylene guanidine hydrochloride antibacterial dressing to accelerate wound healing for infectious skin repair. Ann Transl Med. 2021 Mar;9(6):482.
  • Ni Y, Qian Z, Yin Y, Yuan W, Wu F, Jin T. Polyvinyl Alcohol/Chitosan/Polyhexamethylene Biguanide Phase Separation System: A Potential Topical Antibacterial Formulation with Enhanced Antimicrobial Effect. Molecules. 2020 Mar 15;25(6):1334.
  • Chen J et al. Hydrogen-Bond Assembly of Poly(vinyl alcohol) and Polyhexamethylene Guanidine for Nonleaching and Transparent Antimicrobial Films. CS Appl. Mater. Interfaces2018, 10, 43, 37535–37543
  • English including grammar, style and syntax, should be improved through the professional help from English Editing Company for Scientific Writings.

Author Response

(The authors gave the same response as above.)

Reviewer 4 Report

The authors present a work on "Imaging-Guided Delivery of a Hydrophilic Drug to Eukaryotic Cells Based on its Hydrophobic Ion Pairing with Poly(hexamethylene Guanidine) in a Maleated Chitosan
Carrier" The work is interesting and it expands the interest of  drug delivery applications. However, the work requires further implementation before it can be considered for publication.

1- Keywords should be listed in alphabetical order.

2- The novelty of the work should be presented in the introduction section to differentiate the findings in comparison with previous similar works. 

3- Generalized graphical abstract (If possible) of the work will be highly attractive for reader. 

4- the discussion of each section of the results must be enriched and compared with similar works.

5- Overall resolution of the figures must be improved.

6- Figure 4 "the Fluorescence spectra of aggregates" should be re-blotted and presented differently, the current form is not acceptable.

7- The FTIR in Figure 7 also compact and exhaustive, it can be enhanced to look better. 

8- Conclusion section should be added, containing the main point and summary the finding obtained from the current work. 

Author Response

(The authors gave the same response as above.)

Round 2

Reviewer 3 Report

The manuscript has been sufficiently improved to be published in Molecules journal.